# Delving into female breast cancer: Distinct disease-specific survival outcomes between invasive lobular and ductal carcinomas revealed by propensity score matching

**Wu Zhang[1], Yuquan Huang[2], Ye Zhou[3], Jiaojiao Xue[4], Shan Gao[5], Lin Kang[6], Jian Shi[7], Tao Zhou[8], Yalong Duan[9], Sihan Guo[10], Qingxia Li[3]***

1 Graduate School, North China University of Science and Technology, Tangshan, Hebei, China, 2 Guilin Medical University, Guilin, Guangxi, China, 3 The Fourth Department of Oncology, Hebei General Hospital, Shijiazhuang, China, 4 College of Postgraduate, Hebei Medical University, Shijiazhuang, China, 5 Department of Gland Surgery, Hebei General Hospital, Shijiazhuang, China, 6 Department of Pathology, Hebei General Hospital, Shijiazhuang, China, 7 Department of Oncology, The Fourth Hospital of Hebei Medical University, Shijiazhuang, China, 8 Department of Breast Cancer Center, The Fourth Hospital of Hebei Medical University, Shijiazhuang, China, 9 School of Graduate Studies, Hebei North University, Zhangjiakou, China, 10 Department of Computer Science, Durham University, Durham, United Kingdom

* lqx73@163.com

## Abstract

### Purpose

The difference in prognosis between invasive lobular carcinoma (ILC) and invasive ductal carcinoma (IDC) is still controversial in the academic community. Resolving this controversy can help to more accurately determine patients' prognosis, provide further personalized treatment, alleviate unnecessary psychological burden for some patients, and provide direction for further fundamental research.

### Patients and methods

A retrospective cohort study was conducted using the SEER Research Plus Data 8 Registries, Nov 2021 sub (1978–2019), including female breast cancer patients diagnosed with ILC or IDC between 2010 and 2015. Univariate and multivariate Cox regression analyses were performed, and key covariates affecting prognosis were selected. Propensity score matching (PSM) was employed to match patients, and balance tests were conducted to evaluate covariate distribution. Disease-specific survival (DSS) differences between the matched IDC and ILC groups were compared.

### Results

Following PSM, the covariate differences between the IDC and ILC groups were significantly reduced. The survival analysis revealed a significantly better prognosis for the IDC group than the ILC group (Log-rank test p < 0.001), with a 28.0% increased risk observed in the ILC group.

**Data Availability Statement:** All rowdata files are available from the Seer database (https://seer.cancer.gov/registries/cancer_registry/index.html).

**Funding:** This research was supported by the Precision Medicine Joint Cultivation Program of the Natural Science Foundation of Hebei Province (No. H2022307024). Funders have no role in study design, data collection and analysis, publication decisions, or manuscript preparation.

**Competing interests:** The authors have declared that no competing interests exist.

## Conclusion

This study provides evidence supporting the existence of significant differences in prognosis between IDC and ILC patients after rigorous matching. The IDC group displayed a significantly better prognosis than the ILC group. Notably, these findings have implications for personalized treatment in clinical practice and contribute to the ongoing academic debate on survival differences between IDC and ILC. However, further research is needed to investigate the biological mechanisms, gene expression, and signaling pathway disparities between IDC and ILC, aiming to provide more targeted guidance for clinical decision-making.

## Introduction

According to GLOBOCAN 2020 estimates by the International Agency for Research on Cancer, female breast cancer has surpassed lung cancer as the most commonly diagnosed cancer [1]. With the incidence of breast cancer increasing yearly, it has become one of the significant threats to women's health [2]. Invasive lobular carcinoma (ILC) and invasive ductal carcinoma (IDC) are the two most common histological types of breast cancer, with ILC accounting for 10–15% of all breast cancers and IDC accounting for 70–80% of female breast tumors. Despite sharing similar etiologies, there are significant differences in biological characteristics, clinical manifestations, and prognosis between ILC and IDC [3–5]. Thus, investigating the differences in Disease-Specific Survival (DSS) among different types of breast cancer patients holds great significance.

There is currently a debate in the academic community regarding the prognostic differences between ILC and IDC [6]. In this study, we used propensity score matching (PSM) based on generalized linear models to analyze the differences in DSS rates between these two types of breast cancer patients. By comparing and balancing both patient groups' clinical characteristics, molecular subtypes, and treatment methods, we aim to reveal potential prognostic differences and provide more accurate prognostic evaluations and treatment recommendations.

PSM is a widely used statistical method in observational studies to eliminate differences between two groups of patients with varying clinical and pathological characteristics, making the study results more reliable [7, 8]. Additionally, the PSM method based on generalized linear models can adjust for patient background variables, achieving a balance between comparison groups and reducing bias caused by confounding factors [9, 10]. Through an in-depth analysis of the prognostic differences between ILC and IDC, we aim to provide new evidence to resolve this controversy and bring new insights to clinical practice, ultimately improving breast cancer patients' survival and quality of life.

## Material and methods

### Study design and population

The data used in this study were obtained from the Surveillance, Epidemiology, and End Results (SEER) program, the primary program employed by the National Cancer Institute (NCI) to support cancer surveillance activities in the United States. The SEER program is an authoritative source of information on cancer incidence and survival [11]. It collects and publishes cancer incidence and survival data from population-based cancer registries, covering

approximately 48 percent of the U.S. population [https://seer.cancer.gov/registries/cancer_registry/index.html].

After obtaining research permission (Sequence Number: 24787-Nov2021), we utilized SEER*Stat version 8.4.0.1 to retrieve the SEER Research Plus Data 8 Registries. Nov 2021 sub (1978–2019) contains 4,765,822 cases, of which 4,351,209 are malignant cases. Since the patient data provided by SEER is anonymized, no identifiable information is included in our analysis. Therefore, informed consent was not required.

All female patients included in our analysis had primary tumors in the breast and were diagnosed with positive histology, confirmed by diagnostic confirmation. The sources of the reports included hospital inpatient, outpatient or clinic, radiation therapy or oncology center (2006 and later), hospital or private laboratory, physician's office/private practice (LMD), nursing/convalescent home/hospice, and other hospital outpatient departments or surgical centers (2006 and later), excluding cases obtained from autopsies and those with only death certificate records. The cause of death (COD) and follow-up/survival months were available as complete dates, and samples with unknown causes of death and follow-up time were excluded. Furthermore, COD to site recode included alive and breast cancer-specific death patients. We also screened breast cancer patients with ICD-O-3 codes 8500/3: invasive ductal carcinoma and 8520/3: invasive lobular carcinoma.

Additionally, we obtained patients' age (<60 years, > = 60 years), race (White, Black, Other—including Chinese, Japanese, Vietnamese, Korean, and other Asian ethnicities), marital status (Married, Divorced, Widowed, Single), primary site (Central, Upper Inner, Lower Inner, Upper Outer, Lower Outer, Overlapping, Others), grade (I, II, III, IV), summary stage (Localized, Regional, Distant), AJCC stage (I, II, III, IV), laterality (Right, Left), surgical and radiation sequence (including PORT, NROS, others), systemic surgery sequence (including AAT, NSOST, others), ER/PR status (positive, negative;If 1% or more cells stain positive, the test results are considered positive; if less than 1% of cells stain positive, the results are considered negative), tumor size (< = 1cm, < = 2cm, < = 3cm, < = 4cm, < = 5cm, >5cm), and subtype (HR+/HER2+, HR+/HER2-, HR-/HER2+, HR-/HER2-) from the SEER database. Overall, a total of 66,405 patient samples were collected for our analysis.

## Statistical analysis

We divided the dataset into two groups based on histology: the IDC group and the ILC group. To ensure the accuracy and reliability of the analysis, we first used the ks.test function in R version 4.2.1 to perform a Kolmogorov-Smirnov (KS) test on the only continuous variable-survival to select an appropriate inter-group comparison analysis method. Although the Shapiro-Wilk test is considered a better normality test method, it has higher sensitivity to small sample sizes. When the sample size is large, the Shapiro-Wilk test may result in a misjudgment of P-values [12]. Thus, the KS test is more suitable [13]. Based on the results of the KS test (D = 0.053628, p-value < 0.001), we compared the between-group differences in continuous non-normally distributed variables using the Mann-Whitney U test. Then, we described the sample distribution characteristics of this type of variable using the median (interquartile range) [14]. We used the chi-square test for other categorical variables, which is suitable for large sample sizes, to compare the distribution differences between the IDC and ILC groups. Moreover, we calculated the corresponding P-values and described the distribution of samples within each group using frequency (percentage) [15]. Through this method, we could fully understand the differences in various variables between the IDC and ILC groups, providing a strong basis for the subsequent analysis using the PSM method based on generalized linear models.

Before PSM, we considered the presence of confounding factors. To reduce bias risks and achieve better-matching results, we identified important variables affecting prognosis through Cox regression analysis. We included them as input variables in the PSM model. Specifically, we used the coxph function of the survival package to perform univariate Cox regression analysis for the IDC and ILC groups, respectively, and included covariates with significant statistical differences (P < 0.050) in subsequent multivariate Cox regression analysis. Based on the potential interaction between covariates, we further estimated the independent prognostic ability of each variable. We then evaluated the relationship between each variable and survival by calculating the hazard ratio, 95% confidence interval, and P-value for both univariate and multivariate variable analyses. Furthermore, we excluded variables with P-values greater than 0.050 in univariate and multivariate Cox regression analysis results for the IDC and ILC groups. We also used the remaining variables as covariates closely related to survival for subsequent PSM to minimize the weakening of validity and unnecessary sample loss brought about by including too many irrelevant variables.

Afterward, we constructed a binary logistic regression model with histology as the dependent variable and the covariates identified from the Cox regression analysis as independent variables using the generalized linear model (GLM) function of the matchit package. The propensity score value, representing the probability of each individual being assigned to the IDC or ILC group, was calculated for each individual. Then, using the nearest neighbor matching algorithm with a caliper matching threshold of 0.0001 and a 1:1 ratio, we matched IDC and ILC individuals based on their propensity scores and randomized the matching order to improve matching precision and reduce confounding bias. Thus, this improves the quality of comparing the two pathological types regarding the difference in prognosis and risk. To further screen the variables that had a more significant impact on the grouping, we subsequently used the love.plot function of the cobalt package to conduct a balance test after matching, setting a threshold of 0.05. We repeated the comparison of differences between different variables in the IDC and ILC groups. After matching, we plotted the baseline table of clinical data to verify whether the differences between the groups were effectively controlled and ensure the reliability of the study results. Finally, we compared the prognosis differences between the matched IDC and ILC groups using the survival function of the R package survival. Furthermore, we validated the significance of the differences in prognosis between the two groups using three statistical methods: the log-rank test, the Wald test, and the likelihood ratio test.

## Results

We obtained 510,385 primary breast cancer samples from the SEER Research Plus Data 8 Registries. Nov 2021 sub (1978–2019), of which 66,405 female breast cancer patients were included in our analysis after excluding and including samples according to our criteria. Of these, 58,497 cases were IDC (88.1%), and 7,908 were ILC (11.9%). Additionally, there were 5,726 cases of breast cancer-specific deaths (9.9%) in the IDC group and 853 cases (10.8%) in the ILC group. The median survival time in the IDC group was 75.0 (58.0, 95.0) months, while in the ILC group, it was 73.0 (57.0, 93.0) months. Moreover, the distributions of age, race, marital status, primary tumor site, molecular subtype, pathological grade, summary stage, AJCC, order of surgery and systemic therapy, order of surgery and radiation therapy, hormone status, and tumor size were statistically significantly different between the IDC and ILC groups. Notably, the proportion of elderly patients in the ILC group (61.1%) was significantly higher than that in the IDC group (50.8%). Regarding molecular subtypes, the HR+/HER2- subtype accounted for 94.0% of the ILC group. In contrast, it was only 72.0% in the IDC group, and the other three subtypes were lower in proportion in the ILC group than in the IDC group. In terms of pathological grade, the

proportion of Grade I (31.6%) and Grade II (61.3%) in the ILC group was significantly higher than that in the IDC group (23.4% and 41.5%, respectively). In comparison, the proportion of Grade III (35.0%) and Grade IV (0.1%) in the IDC group was significantly higher than that in the ILC group (Grade III: 7.1%, Grade IV: 0.0%). Furthermore, the proportions of ER-positive (98.1%) and PR-positive (83.5%) patients in the ILC group were significantly higher than those in the IDC group (ER: 82.2%, PR: 71.8%) in terms of hormone status. For tumor size, the proportion of tumors with a diameter of 2cm or less was higher in the IDC group than in the ILC group (64.8% vs. 51.0%). In comparison, the proportion of tumors with a diameter of 2cm or more in the ILC group was higher than that in the IDC group. (Table 1).

The univariate and multivariate Cox regression analysis results of IDC and ILC patients are summarized in S1 and S2 Tables. From the univariate Cox regression analysis of IDC, laterality showed no significant independent prognostic ability (P = 0.747), and the independent prognostic efficacy of the other subcategories in the surgery and radiation therapy sequence was insignificant (P = 0.143). From the multivariate Cox regression analysis, none of the seven Primary_site prognostic abilities were significant, and the prognostic ability of the HR-/HER2- subtype was insignificant (P = 0.743). The prognostic efficacy of the other subcategories in the surgery and radiation therapy sequence was also insignificant (P = 0.260) (S1 Table).

From the univariate Cox regression analysis of ILC, the independent prognostic efficacy of the other subcategories in Primary_site was insignificant (P = 0.263). Laterality showed no significant independent prognostic ability (P = 0.633), and the independent prognostic efficacy of the HR+/HER2- subtype was insignificant (P = 0.118). From the multivariate Cox regression analysis of ILC, Black patients and others showed no significant prognostic ability compared to White patients. The single marital status showed no significant prognostic efficacy (P = 0.490). Additionally, none of the seven different Primary site prognostic abilities were significant, consistent with the multiple regression analysis of IDC. The prognostic efficacy of the other subcategories in the surgery and radiation therapy sequence in ILC was insignificant (P = 0.780), and the prognostic efficacy of the HR+/HER2- subtype was insignificant (P = 0.199). Furthermore, tumor size between 2 and 3 cm and greater than 5 cm did not show a clear prognostic association compared to a tumor size of 1 cm (S2 Table).

The prognostic ability of Grade IV was insignificant in both univariate and multivariate Cox regression analysis. Still, we consider this to be due to the small number of Grade IV samples in ILC rather than its true situation. Finally, we included variables other than laterality that had no significant prognostic ability in both univariate and multivariate Cox regression results in the calculation of the propensity score.

We first calculated the absolute standardized mean difference (AMD) of each covariate from the balance test of covariates before and after PSM. Notably, AMD is a measure for evaluating the degree of distributional differences of covariates before and after matching [16], and the absolute mean difference graph of covariates was plotted (Fig 1, left panel). It can be seen from the graph that the absolute standardized mean differences between each covariate of IDC and ILC groups were all less than 0.050 after PSM, suggesting that differences between each covariate in the two groups were significantly reduced after matching. Moreover, in the baseline data distribution after PSM (Table 2), there were no significant differences observed in the distribution of all variables related to DSS between IDC and ILC groups (P>0.050). However, the absolute standardized mean differences of covariates such as age, AJCC stage, grade classification, ER and PR status, molecular subtypes, and tumor size were all greater than 0.05 before PSM, indicating significant differences in these covariates' distribution.

Next, we analyzed the differences between each covariate in the cumulative distribution functions and applied the KS statistic. The KS statistic is a non-parametric test method to measure the difference between two distributions. It is calculated as the maximum distance

**Table 1. Clinical data baseline table before propensity score matching.**

| level | | Overall | IDC | ILC | p |
|---|---|---|---|---|---|
| n | | 66405 | 58497 | 7908 | |
| survival (median [IQR]) | | 75.0 [57.0, 95.0] | 75.0 [58.0, 95.0] | 73.0 [57.0, 93.0] | <0.001 |
| status (%) | | | | | |
| | Alive | 59826 (90.1) | 52771 (90.2) | 7055 (89.2) | 0.006 |
| | Dead | 6579 (9.9) | 5726 (9.8) | 853 (10.8) | |
| age (%) | | | | | |
| | <60 | 31885 (48.0) | 28806 (49.2) | 3079 (38.9) | <0.001 |
| | > = 60 | 34520 (52.0) | 29691 (50.8) | 4829 (61.1) | |
| race (%) | | | | | |
| | White | 55083 (83.0) | 48103 (82.2) | 6980 (88.3) | <0.001 |
| | Black | 5879 (8.9) | 5339 (9.1) | 540 (6.8) | |
| | Other | 5443 (8.2) | 5055 (8.6) | 388 (4.9) | |
| Maritial status (%) | | | | | |
| | Married | 40847 (61.5) | 35952 (61.5) | 4895 (61.9) | <0.001 |
| | Divorced | 7807 (11.8) | 6880 (11.8) | 927 (11.7) | |
| | Widowed | 7308 (11.0) | 6299 (10.8) | 1009 (12.8) | |
| | single | 10443 (15.7) | 9366 (16.0) | 1077 (13.6) | |
| Primary Site (%) | | | | | |
| | Central | 2755 (4.2) | 2376 (4.1) | 379 (4.8) | <0.001 |
| | Upper inner | 8704 (13.1) | 7869 (13.5) | 835 (10.6) | |
| | Lower inner | 3787 (5.7) | 3466 (5.9) | 321 (4.1) | |
| | Upper outer | 23034 (34.7) | 20256 (34.6) | 2778 (35.1) | |
| | Lower outer | 5270 (7.9) | 4659 (8.0) | 611 (7.7) | |
| | Overlapping | 15453 (23.3) | 13621 (23.3) | 1832 (23.2) | |
| | others | 7402 (11.2) | 6250 (10.7) | 1152 (14.6) | |
| Subtype (%) | | | | | |
| | HR+/HER2+ | 7020 (10.6) | 6687 (11.4) | 333 (4.2) | <0.001 |
| | HR+/HER2- | 49513 (74.6) | 42081 (72.0) | 7432 (94.0) | |
| | HR-/HER2+ | 2844 (4.3) | 2807 (4.8) | 37 (0.5) | |
| | HR-/HER2- | 7028 (10.6) | 6922 (11.8) | 106 (1.3) | |
| Grade (%) | | | | | |
| | I | 16173 (24.4) | 13678 (23.4) | 2495 (31.6) | <0.001 |
| | II | 29097 (43.8) | 24247 (41.5) | 4850 (61.3) | |
| | III | 21054 (31.7) | 20494 (35.0) | 560 (7.1) | |
| | IV[(1)] | 81 (0.1) | 78 (0.1) | 3 (0.0) | |
| Summary stage (%) | | | | | |
| | Localized | 44707 (67.3) | 39655 (67.8) | 5052 (63.9) | <0.001 |
| | Regional | 19241 (29.0) | 16713 (28.6) | 2528 (32.0) | |
| | Distant | 2457 (3.7) | 2129 (3.6) | 328 (4.2) | |
| AJCC (%) | | | | | |
| | I | 36096 (54.4) | 32596 (55.7) | 3500 (44.3) | <0.001 |
| | II | 21498 (32.4) | 18585 (31.8) | 2913 (36.8) | |
| | III | 6458 (9.7) | 5280 (9.0) | 1178 (14.9) | |
| | IV | 2353 (3.5) | 2036 (3.5) | 317 (4.0) | |
| Laterality (%) | | | | | |
| | Right | 32886 (49.5) | 29037 (49.6) | 3849 (48.7) | 0.109 |
| | Left | 33519 (50.5) | 29460 (50.4) | 4059 (51.3) | |

*(Continued)*

**Table 1.** (Continued)

| level | | Overall | IDC | ILC | p |
|---|---|---|---|---|---|
| Systemic Sur Seq (%) | | | | | |
| | AAT | 45514 (68.5) | 39718 (67.9) | 5796 (73.3) | <0.001 |
| | NSOST | 13284 (20.0) | 11795 (20.2) | 1489 (18.8) | |
| | others | 7607 (11.5) | 6984 (11.9) | 623 (7.9) | |
| Surg Rad Seq (%) | | | | | |
| | PORT | 37712 (56.8) | 33461 (57.2) | 4251 (53.8) | <0.001 |
| | NROS | 27921 (42.1) | 24319 (41.6) | 3602 (45.6) | |
| | others | 772 (1.2) | 717 (1.2) | 55 (0.7) | |
| ER (%) | | | | | |
| | Positive | 55861 (84.1) | 48107 (82.2) | 7754 (98.1) | <0.001 |
| | Negtive | 10544 (15.9) | 10390 (17.8) | 154 (2.0) | |
| PR (%) | | | | | |
| | Positive | 48625 (73.2) | 42019 (71.8) | 6606 (83.5) | <0.001 |
| | Negtive | 17780 (26.8) | 16478 (28.2) | 1302 (16.5) | |
| Tumor size (%) | | | | | |
| | < = 1 | 18397 (27.7) | 16822 (28.8) | 1575 (20.0) | <0.001 |
| | < = 2 | 23557 (35.5) | 21096 (36.1) | 2461 (31.1) | |
| | < = 3 | 12165 (18.3) | 10656 (18.2) | 1509 (19.1) | |
| | < = 4 | 5000 (7.5) | 4320 (7.4) | 680 (8.6) | |
| | < = 5 | 2559 (3.9) | 2092 (3.6) | 467 (5.9) | |
| | >5 | 4727 (7.1) | 3511 (6.0) | 1216 (15.4) | |

**Abbreviation:** [1]including undifferentiated; anaplastic; Grade IV

AAT, Adjuvant Therapy; NSOST, No systemic therapy and/or surgical therapy; PORT, Post-Operative Radiation Therapy; NROS, No radiation and/or cancer-directed surgery.

between the cumulative distribution functions of the two distributions [17]. In this study, we calculated the KS statistic for each covariate before and after matching (Fig 1, right panel). The KS statistic of covariates such as age, AJCC stage, grade classification, ER and PR status, molecular subtypes, and tumor size were all greater than 0.05 before PSM, indicating that the distribution of these covariates within the sample before matching was unbalanced. However, after PSM, the KS statistic of each covariate was less than 0.05, almost approaching 0, consistent with the distribution of absolute mean difference, implying that the sample was balanced on these covariates after matching.

Finally, we conducted a survival analysis on the matched IDC and ILC groups. The KM survival analysis results showed that the prognosis of the IDC group was significantly better than that of the ILC group (Log-rank test p < 0.001) (Fig 2). Compared with the IDC group, Cox proportional hazards regression analysis showed that the risk of the ILC group was 28.0% higher than the IDC group (HR = 1.28, CI = 1.12–1.45). Meanwhile, the results of the likelihood ratio test (W value: 1.96, p < 0.001) and Wald test (W value: 11.87, p = < 0.001) were consistent with the Log-rank test results, further confirming the significant difference in prognosis between the IDC and ILC groups.

## Discussion

IDC is the most prevalent breast cancer type, believed to primarily originate from malignant epithelial cells of the breast's terminal ducts [18]. IDC is characterized by invasive growth, with

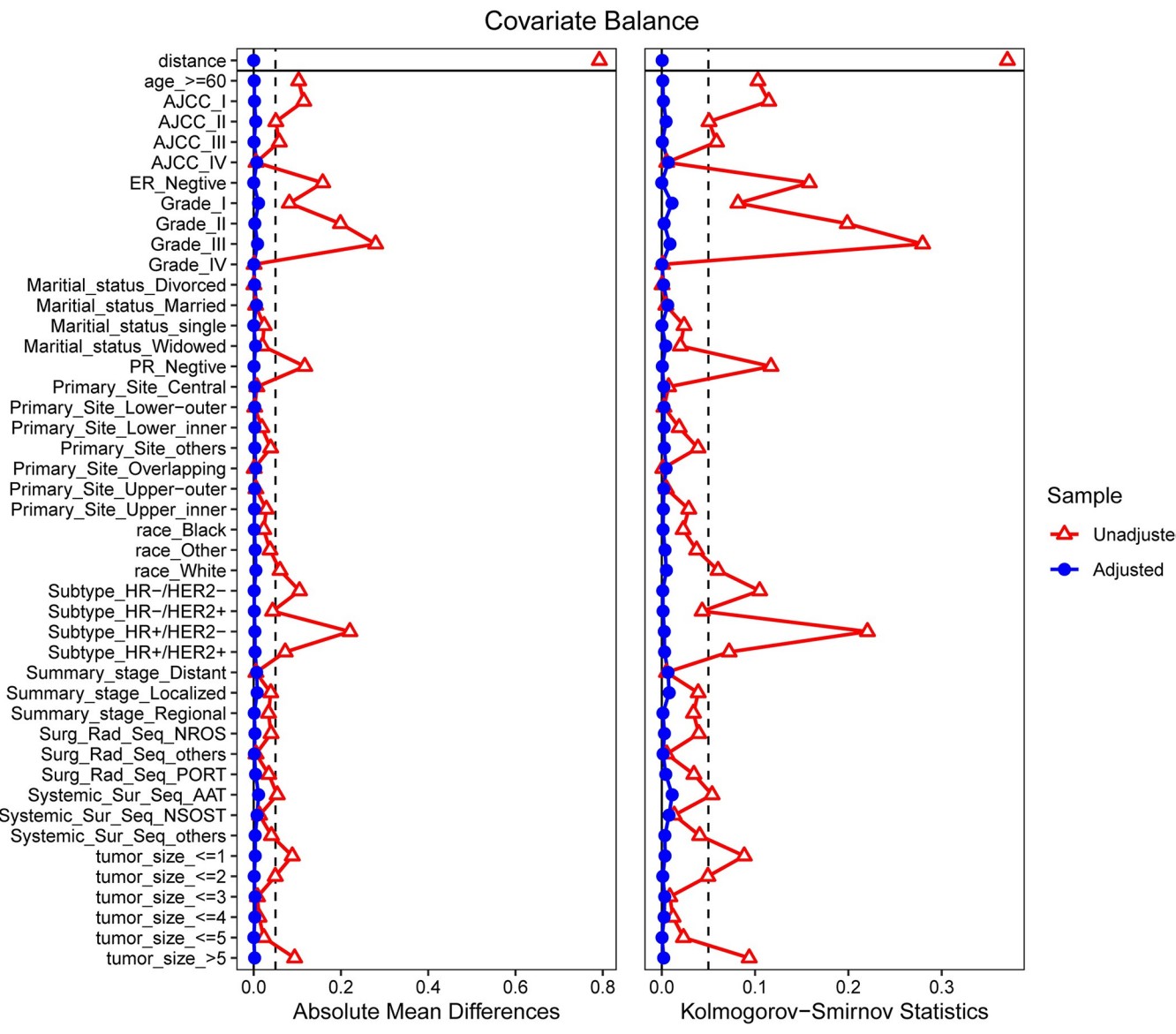

**Fig 1. Covariate balance test plot after propensity score matching.**

tumor cells infiltrating the breast stroma and spreading after breaking through ductal walls. Microscopically, tumor cells often form cord-like, cluster-like, or small beam-like patterns. Clinically, IDC manifests mainly as a nipple-like and solid mass. IILC, the second most common invasive breast cancer histological type after IDC, originates primarily from malignant cells beneath the breast lobule's epithelium [19]. Microscopically, ILC often exhibits small, uniform, and non-adhesive cancer cells distributed individually or diffusely in the fibrous stroma [20]. Compared to IDC, ILC lacks apparent clinical masses and has a lower detection rate on breast ultrasound and X-ray examinations [21].

Besides the aforementioned histological differences, ILC and IDC also display molecular disparities. ILC patients lack E-cadherin expression, a deficiency that stimulates the activation signal of Epithelial-Mesenchymal Transition (EMT), resulting in a higher incidence of bone, gastrointestinal, omental, and ovarian metastases compared to IDC patients [22–25].

**Table 2. Baseline clinical data after propensity score matching.**

| Characteristic | | Overall | IDC | ILC | p |
|---|---|---|---|---|---|
| NO. | | 12794 | 6397 | 6397 | |
| survival (median [IQR]) | | 75.000 [58.0, 95.0] | 76.000 [59.0, 96.0] | 75.000 [58.0, 94.0] | 0.005 |
| status (%) | | | | | |
| | Alive | 11821 (92.4) | 5944 (92.9) | 5877 (91.9) | 0.028 |
| | Dead | 973 (7.6) | 453 (7.1) | 520 (8.1) | |
| age (%) | | | | | |
| | <60 | 5216 (40.8) | 2625 (41.0) | 2591 (40.5) | 0.553 |
| | > = 60 | 7578 (59.2) | 3772 (59.0) | 3806 (59.5) | |
| race (%) | | | | | |
| | Black | 690 (5.4) | 344 (5.4) | 346 (5.4) | 0.976 |
| | Other | 633 (5.0) | 314 (4.9) | 319 (5.0) | |
| | White | 11471 (89.6) | 5739 (89.7) | 5732 (89.6) | |
| Maritial status (%) | | | | | |
| | Divorced | 1385 (10.8) | 699 (10.9) | 686 (10.7) | 0.979 |
| | Married | 8283 (64.7) | 4140 (64.7) | 4143 (64.8) | |
| | single | 1654 (12.9) | 827 (12.9) | 827 (12.9) | |
| | Widowed | 1472 (11.5) | 731 (11.4) | 741 (11.6) | |
| Primary Site (%) | | | | | |
| | Central | 516 (4.0) | 261 (4.1) | 255 (4.0) | 0.999 |
| | Lower outer | 962 (7.5) | 481 (7.5) | 481 (7.5) | |
| | Lower inner | 508 (4.0) | 256 (4.00) | 252 (3.9) | |
| | others | 1675 (13.1) | 826 (12.9) | 849 (13.3) | |
| | Overlapping | 2996 (23.4) | 1502 (23.5) | 1494 (23.4) | |
| | Upper outer | 4733 (37.0) | 2368 (37.0) | 2365 (37.0) | |
| | Upper inner | 1404 (11.0) | 703 (11.0) | 701 (11.0) | |
| Subtype (%) | | | | | |
| | HR-/HER2- | 181 (1.4) | 85 (1.3) | 96 (1.5) | 0.711 |
| | HR-/HER2+ | 66 (0.5) | 34 (0.5) | 32 (0.5) | |
| | HR+/HER2- | 11984 (93.7) | 5987 (93.6) | 5997 (93.8) | |
| | HR+/HER2+ | 563 (4.4) | 291 (4.6) | 272 (4.3) | |
| Grade (%) | | | | | |
| | I | 3725 (29.1) | 1833 (28.7) | 1892 (29.6) | 0.412 |
| | II | 7977 (62.4) | 4004 (62.6) | 3973 (62.1) | |
| | III | 1092 (8.5) | 560 (8.8) | 532 (8.3) | |
| AJCC (%) | | | | | |
| | I | 6833 (53.4) | 3427 (53.6) | 3406 (53.2) | 0.400 |
| | II | 4354 (34.0) | 2186 (34.2) | 2168 (33.9) | |
| | III | 1279 (10.0) | 635 (9.9) | 644 (10.1) | |
| | IV | 328 (2. 6) | 149 (2.3) | 179 (2.8) | |
| Laterality (%) | | | | | |
| | Left | 6372 (49.8) | 3066 (47.9) | 3306 (51.7) | <0.001 |
| | Right | 6422 (50.2) | 3331 (52.1) | 3091 (48.3) | |
| Systemic Sur Seq (%) | | | | | |
| | AAT | 9643 (75.4) | 4846 (75.8) | 4797 (75.0) | 0.588 |
| | NSOST | 2402 (18.8) | 1185 (18.5) | 1217 (19.0) | |
| | others | 749 (5.9) | 366 (5.7) | 383 (6.0) | |
| Surg Rad Seq (%) | | | | | |

*(Continued)*

**Table 2.** (Continued)

| Characteristic | | Overall | IDC | ILC | p |
|---|---|---|---|---|---|
| | NROS | 5732 (44.8) | 2860 (44.7) | 2872 (44.9) | 0.541 |
| | others | 71 (0.6) | 31 (0.5) | 40 (0.6) | |
| | PORT | 6991 (54.6) | 3506 (54.8) | 3485 (54.5) | |
| ER (%) | | | | | |
| | Negtive | 266 (2.1) | 127 (2.0) | 139 (2.2) | 0.496 |
| | Positive | 12528 (97.9) | 6270 (98.0) | 6258 (97.8) | |
| PR (%) | | | | | |
| | Negtive | 1669 (13.6) | 836 (13.1) | 833 (13.0) | 0.958 |
| | Positive | 11125 (87.0) | 5561 (86.9) | 5564 (87.0) | |
| Tumor size (%) | | | | | |
| | < = 1 | 3041 (23.8) | 1514 (23.7) | 1527 (23.9) | 0.937 |
| | < = 2 | 4655 (36.4) | 2317 (36.2) | 2338 (36.6) | |
| | < = 3 | 2622 (20.5) | 1323 (20.7) | 1299 (20.3) | |
| | < = 4 | 944 (7.4) | 464 (7.3) | 480 (7.5) | |
| | < = 5 | 470 (3.7) | 243 (3.8) | 227 (3.6) | |
| | >5 | 1062 (8.3) | 536 (8.4) | 526 (8.2) | |

**Abbreviation:** AAT, Adjuvant Therapy; NSOST, No systemic therapy and/or surgical therapy; PORT, Post-Operative Radiation Therapy; NROS, No radiation and/or cancer-directed surgery.

Moreover, studies have revealed that genomic profiles of ILC and IDC patients differ based on tissue sequencing assessments. ILC patients exhibit a higher tumor mutation burden and variations in immune infiltration and copy number changes [26–29]. Furthermore, even though the clinical treatment methods for both are relatively similar, some differences still exist. For instance, in clinical practice, the frequency of ILC patients undergoing mastectomy is slightly higher than that of IDC patients [24]. Furthermore, there is ongoing debate regarding the clinical outcomes of ILC and IDC patients. Wasif et al. discovered that ILC patients had a significantly higher five-year DSS rate compared to IDC patients (P < 0.001) through stage-matched analysis [30]. Chen et al. obtained similar results, with ILC patients having a survival advantage within five years post-surgery compared to IDC patients but demonstrating a worse long-term prognosis [24]. Furthermore, Yang et al. found no significant difference in overall survival rates between ILC and IDC patients during the first five years after PMS analysis [31]. In contrast, some studies have reported that ILC patients have a higher ten-year overall survival rate than IDC patients [32, 33].

Considering the abovementioned research, we hypothesize that the differences in study outcomes may stem from variations in the chosen study populations, variable control, and distinct data analysis methods. Therefore, to provide more comprehensive and precise data, this study matched with a more comprehensive set of covariates and their subclasses, eliminating the impact of baseline patient characteristics. Using PSM based on a generalized linear model, we explored the disparities in DSS between these two types of breast cancer patients. Through balancing tests of covariates before and after matching, the disparities in various covariates between the IDC and ILC groups were significantly reduced, offering a more balanced foundation for subsequent survival analysis. From the survival analysis post-PSM, we discovered that the prognosis of the IDC group was significantly better than that of the ILC group (Log-rank test p < 0.001), and the risk of the ILC group increased by 28.0% compared to the IDC group. This result indicates that after PSM, the difference in tumor-specific survival between IDC and

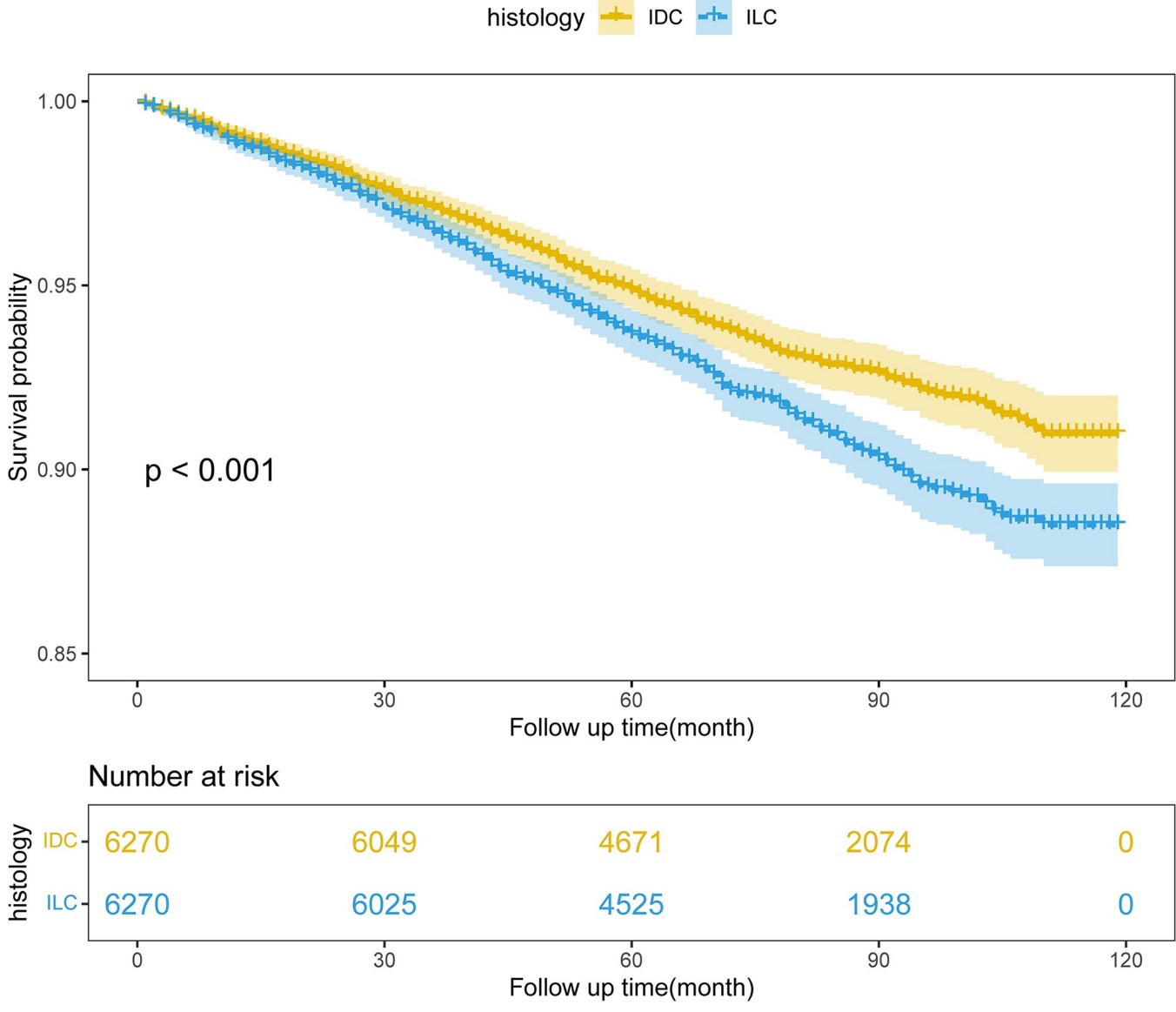

**Fig 2. KM survival analysis of IDC and ILC groups after PSM.**

ILC patients is substantial, supporting the notion that IDC and ILC have distinct clinical outcomes. In contrast, a recent study conducted by Ciqiu Yang et al. showed that prior to PSM, the overall survival (OS) of IDC patients was superior to that of ILC patients (HR = 1.045, P = 0.025, 95%CI: 1.007–1.085) [31]. However, after PSM, the difference in OS between the two groups disappeared. In addition to the reasons for these divergent results mentioned above, the differing focus outcomes may be one of the leading causes of such discrepancies. For breast cancer patients with longer survival periods, DSS can help us avoid the influences of other potential interfering factors more effectively than OS, thereby reflecting the impact of breast cancer on survival more accurately.

This study has some noted limitations. As a retrospective study, the data is sourced from the SEER database. Although this is an authoritative large-scale cancer registry database, its population coverage primarily comprises white individuals in the United States. To further

verify the universality and representativeness of these conclusions, it is necessary to expand the sample sources and scope for patients from other regions and ethnicities. Additionally, although we have substantially reduced biases through PSM, there may still be unobserved potential confounding factors that could affect the accuracy of the results.

## Conclusion

Through applying PSM, our study discovered that, upon stringent matching, the prognosis for patients in the IDC group was markedly superior to those in the ILC group. This finding offers compelling evidence to support individualized treatment strategies for patients with IDC and ILC in clinical settings. Nevertheless, it remains essential for future research to delve deeper into the distinctions between IDC and ILC concerning biological mechanisms, gene expression patterns, and signaling pathways to provide more precise and targeted guidance in clinical practice.

## Supporting information

**S1 Table. IDC group univariate and multivariate Cox regression analysis.**
(DOCX)

**S2 Table. ILC group univariate and multivariate Cox regression analysis.**
(DOCX)

**S1 File. Request for change to authorship.**
(DOCX)

## Acknowledgments

We thank all medical students who devoted their time to this study. The authors thank AiMi Academic Services (www.aimieditor.com) for English language editing and review services.

## Author Contributions

**Conceptualization:** Ye Zhou, Tao Zhou, Yalong Duan.

**Data curation:** Wu Zhang, Yuquan Huang.

**Formal analysis:** Ye Zhou, Jiaojiao Xue, Sihan Guo.

**Investigation:** Jiaojiao Xue.

**Methodology:** Jiaojiao Xue, Jian Shi, Tao Zhou, Qingxia Li.

**Resources:** Shan Gao.

**Software:** Yuquan Huang, Shan Gao.

**Supervision:** Ye Zhou, Shan Gao, Jian Shi, Tao Zhou, Qingxia Li.

**Validation:** Jiaojiao Xue, Sihan Guo.

**Visualization:** Lin Kang.

**Writing – original draft:** Wu Zhang, Yalong Duan.

**Writing – review & editing:** Wu Zhang, Yuquan Huang, Sihan Guo, Qingxia Li.

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
