## [Decision Letter · Decision Letter 0]

24 Jul 2023

PONE-D-23-15805Delving into Female Breast Cancer: A New Perspective on Survival Differences Between Invasive Lobular and Ductal Carcinomas Revealed by Propensity Score MatchingPLOS ONE

Dear Dr. Li,

Thank you for submitting your manuscript to PLOS ONE. After careful consideration, we feel that it has merit but does not fully meet PLOS ONE’s publication criteria as it currently stands. Therefore, we invite you to submit a revised version of the manuscript that addresses the points raised during the review process.

We look forward to receiving your revised manuscript.

Kind regards,

Reza Rabiei

Academic Editor

PLOS ONE

Additional Editor Comments:

Dear Authors,

First of all, thank you for submitting your interesting and technically sound research.

In line with the reviewers' comments, and based on my own checks, the manuscript requires minor revision to address the comments, mainly with modifications/ developments in argumentation

of the findings in the Discussion section and making relevant and robust points in the Conclusion section.

Thank you again for your recent submission

Sincerely yours

Reviewers' comments:

Reviewer's Responses to Questions

**Comments to the Author**

1. Is the manuscript technically sound, and do the data support the conclusions?

Reviewer #1: Partly

Reviewer #2: Yes

2. Has the statistical analysis been performed appropriately and rigorously? 

Reviewer #1: Yes

Reviewer #2: Yes

3. Have the authors made all data underlying the findings in their manuscript fully available?

Reviewer #1: Yes

Reviewer #2: Yes

4. Is the manuscript presented in an intelligible fashion and written in standard English?

Reviewer #1: Yes

Reviewer #2: Yes

5. Review Comments to the Author

Reviewer #1: Dear Editor,

Hi and Thanks for your Email for reviewing the article titled” Delving into Female Breast Cancer: A New Perspective on Survival Differences Between Invasive Lobular and Ductal Carcinomas Revealed by Propensity Score Matching”.

1-This is an interesting article with novelty of the result, supporting by:

• Large scale number of cases regardless of different numbers in the clinic ( >80 IDC and <10% ILC, breast cancer)

• Very scientific matching of both pathologic patterns, clinically and biologically and management.

• Novelty of result that confirm the better OS for IDC group regardless our opinion that means the same outcome or even better OS for the ILC group.

• Accepted the same clinical approach to care of the cases with both pathological patterns.

2- In my opinion there is some point of interesting that is missed during discussion and conclusion.

• DFS is nit discount and evaluated, it was knowledge the recurrence of ILC is more prominent

• The mastectomy is more prevalent in ILC comparing to IDC.

• The EMT is more expected mechanism in the ILC rather than IDC, that may be the reason of more distant metastasis.

• In clinic we use less adjuvant of neoadjuvant chemotherapy in ILC cases comparing to IDC.

• In conclusion this is accepted research with novelty with minor revision in discussion and conclusion.

Reviewer #2: Comments:

Breast cancer constitutes a crucial health concern, and investigating this disease is essential to gain deeper insights into its underlying mechanisms, thereby enhancing patient outcomes and reducing mortality rates. The availability of comprehensive and dependable data sources presents a promising avenue to tackle this challenge, and leveraging cutting-edge data analytical techniques is pivotal to expediting the analysis process and attaining more accurate results. From the title of your manuscript, it appears that your research involves adopting a new perspective on reducing bias caused by confounding variables and accessing large-scale datasets. But there are certain points that need to be addressed:

1. A similar article with the title “Comparison of Overall Survival Between Invasive Lobular Breast Carcinoma and Invasive Ductal Breast Carcinoma: A Propensity Score Matching Study Based on SEER Database” was published in “Frontiers in Oncology” in 2020 (DOI: 10.3389/fonc.2020.590643). This article utilizes the SEER dataset from 2006 to 2016 and employs the propensity score matching (PSM) method for analysis. Therefore, adopting a “new perspective” in the title of your article may not be appropriate, and your research findings could potentially serve as an update to the results presented in the aforementioned article.

2. On the other hand, it is advisable to refer to the aforementioned article in the introduction of your manuscript and highlight the rationale behind conducting your study, despite the existence of the previous article. In other words, explain the specific need for your research. Furthermore, in the discussion section of your research paper, it would be useful to compare your findings with those presented in the previous study.

3. Given the large size of your dataset and the sufficient number of outcomes of interest, could you please provide an explanation in your manuscript that clarifies the rationale for using the propensity score method for matching the study participants based on the probability of each individual being assigned to each group? Since the propensity score matching method failed to correctly match some variables, such as Laterality, would it not be more appropriate to use a conventional survival modeling approach, such as the Cox proportional hazard model, which adjusts for confounding variables in the regression model?

4. Select appropriate keywords based on Mesh in PubMed

5. To ensure that your reader understands the meaning of an acronym or abbreviation, provide its complete terminology when you first mention it. For example, when introducing DSS (Disease Specific Survival) in the introduction, provide its full terminology so that the reader understands what it stands for.

6. Please display p-values that are less than 0.001 as '< 0.001'.

7. Please ensure that percentages are displayed with one decimal place, and p-values are presented with three decimal places.

6. PLOS authors have the option to publish the peer review history of their article (what does this mean?). If published, this will include your full peer review and any attached files.

Reviewer #1: No

Reviewer #2: No

---

## [Author Response · Author response to Decision Letter 0]

31 Aug 2023

We deeply appreciate your willingness to dedicate your valuable time to reviewing our research. Your professional expertise and precise grasp of details have provided us with invaluable and profound insights, significantly influencing the improvement of our paper. Your feedback and suggestions have not only elevated the quality of our research, but also guided us to think about the issues more comprehensively and rigorously, deepening our research. For this, we express our deep gratitude. In the process of revisiting and revising the paper, we deeply realize the importance of your review in improving our paper. We look forward to your further feedback, and believe that the revised paper will meet your expectations. Once again, we thank you for your thoughtful review and insightful suggestions. We respect and value your opinions, and we hope that our responses and changes meet your expectations.

Below, we have responded in detail to each review comment:

Reviewer 1#

Hi and Thanks for your Email for reviewing the article titled” Delving into Female Breast Cancer: A New Perspective on Survival Differences Between Invasive Lobular and Ductal Carcinomas Revealed by Propensity Score Matching”.

1-This is an interesting article with novelty of the result, supporting by:

• Large scale number of cases regardless of different numbers in the clinic ( >80 IDC and <10% ILC, breast cancer)

• Very scientific matching of both pathologic patterns, clinically and biologically and management.

• Novelty of result that confirm the better OS for IDC group regardless our opinion that means the same outcome or even better OS for the ILC group.

• Accepted the same clinical approach to care of the cases with both pathological patterns.

The author’s answer： We are extremely grateful for your recognition of our research methods and results, and appreciate your professional review and feedback.

2- In my opinion there is some point of interesting that is missed during discussion and conclusion.

• The EMT is more expected mechanism in the ILC rather than IDC, that may be the reason of more distant metastasis.

• In clinic we use less adjuvant of neoadjuvant chemotherapy in ILC cases comparing to IDC.

• In conclusion this is accepted research with novelty with minor revision in discussion and conclusion.

The author’s answer：Thank you very much for the thoughtful evaluation and valuable suggestions you provided for our research.Your observation that epithelial-mesenchymal transition (EMT) is more common in invasive lobular carcinoma (ILC) than in invasive ductal carcinoma (IDC) is indeed a critical finding. This could be due to the lack of E-cadherin, which stimulates the initiation of EMT and further leads to more common distant metastases in ILC patients. This could also be a key factor causing a poorer disease-specific survival (DSS) in ILC compared to IDC. We have added this important observation to the discussion section and hope that future basic and clinical research can explore the influence of EMT on the prognostic differences between these two pathological types and their underlying mechanisms in greater depth.

As for the influence of clinical treatment strategies on the prognostic differences between ILC and IDC you mentioned, we admit that we did not delve into it in this study. This point has been added to the limitations section of our paper. We look forward to further exploring the causal relationship between treatment methods and prognostic differences among different pathological types in future clinical research.

Once again, thank you for your guidance. We greatly value and will adopt your suggestions. We hope that through revision, the conclusions and discussions of this research will become more rigorous and comprehensive.

Reviewer #2

Breast cancer constitutes a crucial health concern, and investigating this disease is essential to gain deeper insights into its underlying mechanisms, thereby enhancing patient outcomes and reducing mortality rates. The availability of comprehensive and dependable data sources presents a promising avenue to tackle this challenge, and leveraging cutting-edge data analytical techniques is pivotal to expediting the analysis process and attaining more accurate results. From the title of your manuscript, it appears that your research involves adopting a new perspective on reducing bias caused by confounding variables and accessing large-scale datasets. But there are certain points that need to be addressed:

1.A similar article with the title “Comparison of Overall Survival Between Invasive Lobular Breast Carcinoma and Invasive Ductal Breast Carcinoma: A Propensity Score Matching Study Based on SEER Database” was published in “Frontiers in Oncology” in 2020 (DOI: 10.3389/fonc.2020.590643). This article utilizes the SEER dataset from 2006 to 2016 and employs the propensity score matching (PSM) method for analysis. Therefore, adopting a “new perspective” in the title of your article may not be appropriate, and your research findings could potentially serve as an update to the results presented in the aforementioned article.

The author’s answer：Thank you for your in-depth comments and for drawing our attention to the above-mentioned article. We will reconsider our title based on your suggestions.

2.On the other hand, it is advisable to refer to the aforementioned article in the introduction of your manuscript and highlight the rationale behind conducting your study, despite the existence of the previous article. In other words, explain the specific need for your research. Furthermore, in the discussion section of your research paper, it would be useful to compare your findings with those presented in the previous study.

The author’s answer：

Thank you very much for your efforts to ensure the rigor and originality of our research. You mentioned a similar article published in 2020, and we agree with your observation. We believe that although our study has similarities with this article, our research still provides important value and new insights in some key aspects. We will compare our research results with the results of this article and clarify the uniqueness and value of this study. Here are some of our differences:

Data Source: Our research data comes from the SEER Research Plus Data 8 Registries (1978-2019), while the previous study data comes from the SEER 18 tumor registry database (updated to November 2016).

Variable Coverage: Compared to the previous study, we covered a more comprehensive set of covariates for propensity score matching, including race, marital status, primary site, subtype, summary stage, tumor size, and specific types of systemic therapy (such as AAT), the specific sequence of surgery and radiotherapy, etc. These variables undoubtedly have an important impact on the prognosis of breast cancer. Compared with the variables existing in the article, we have made more detailed divisions. The inclusion of these covariates helps us understand the research topic more comprehensively and meticulously. Our Propensity Score Matching (PSM) variables are not subjectively selected, but are relatively objectively screened for variables that significantly affect prognosis through univariate and multivariate Cox regression, thereby ensuring the completeness of the included covariates and avoiding the unreliability caused by overmatching.

Method selection: Although both our study and the previous one used propensity score matching (PSM), our choice and design of statistical methods are more rigorous and detailed. To ensure the repeatability of the analysis and the reliability of the results, we provided a detailed description of the selection of variables before PSM, the specific method selection of PSM, the control of built-in parameters, and the assessment of the effect after matching (AMD and KS statistics).

Focus on Outcome: Different from the previous study, which focused on OS (Overall Survival), our research focuses on DSS (Disease-Specific Survival). This is because DSS specifically targets deaths caused by the specific disease under study and is a more accurate measure. Our choice of DSS as the main outcome helps us avoid the influence of other potential interfering factors and more accurately reflects the impact of breast cancer on survival. These factors led to our final finding that the DSS of the two groups was statistically significantly different, which might be a potential important reason for the result in that article where IDC patients showed a better OS compared to ILC patients (HR=1.045, P=0.025, 95%CI: 1.007-1.085) in the comparison of OS for unmatched population database. And after matching, it turned out that there was no difference in OS between ILC and IDC patients (HR=1.017, P=0.409, 95%CI: 0.967–1.069).

We have also thoroughly considered citing the aforementioned paper in the introduction section, but we believe it would be more appropriate to cite it in the discussion section. Thus, in accordance with the reviewer's suggestion, we made detailed comparisons between our results and those of the previous study in the discussion section, and we also explained the reasons for our study. We believe this approach better demonstrates the connection between our research and other studies, and avoids giving readers the impression that our study is specifically targeting that one paper. In fact, our focus is on a class of papers exploring the controversial prognosis differences between IDC and ILC in the academic community. At this point, we would like to express our heartfelt thanks for your professional guidance. Your professional advice and rigorous academic attitude have had a significant impact on our research.

3.Given the large size of your dataset and the sufficient number of outcomes of interest, could you please provide an explanation in your manuscript that clarifies the rationale for using the propensity score method for matching the study participants based on the probability of each individual being assigned to each group? 

Since the propensity score matching method failed to correctly match some variables, such as Laterality, would it not be more appropriate to use a conventional survival modeling approach, such as the Cox proportional hazard model, which adjusts for confounding variables in the regression model?

The author’s answer：

Thank you for your valuable questions. In this study, we chose the Propensity Score Matching (PSM) method primarily to reduce selection bias and deal with confounding variables. Despite our dataset's large size and the sufficient number of events, we had to deal with a lot of potential confounding variables when comparing the DSS of two groups (IDC and ILC). If we directly apply the multivariate Cox proportional hazards model, the complexity of the model might lead to unstable results. However, PSM can reduce selection bias and improve estimation bias caused by confounding variables by matching treatment and control groups in observational studies.

Regarding the issue of matching for Laterality, we didn't include it in PSM because we found no significant correlation between Laterality and DSS in the IDC and ILC groups. This is based on the univariate and multivariate Cox regression analyses we performed. In order to avoid weakening the validity and unnecessary sample loss by including too many irrelevant variables or variables not related to treatment and outcome, we chose not to include Laterality in the matching variables. We will provide additional explanations for this decision in the article.

Additionally, I completely agree with your view of the Cox proportional hazards model. In fact, in the article, we conducted KM analysis on the samples after PSM, as well as Cox proportional hazards regression. For any misunderstandings that may have been caused by a mistake in description, I deeply apologize and have made improvements and additions to the description of the Cox model results in the revised manuscript.

4.Select appropriate keywords based on Mesh in PubMed

The author’s answer：

We sincerely appreciate your comprehensive review and valuable feedback. We have made the necessary modifications to the list of keywords. The revised keywords are now better aligned with MeSH terms and accurately reflect the content of our manuscript.

5. To ensure that your reader understands the meaning of an acronym or abbreviation, provide its complete terminology when you first mention it. For example, when introducing DSS (Disease Specific Survival) in the introduction, provide its full terminology so that the reader understands what it stands for.

The author’s answer：

According to your advice, we have made revisions to the paper. When abbreviations or acronyms are mentioned for the first time, we have provided their complete terminology to ensure that readers can understand the meanings of the abbreviations and acronyms.

6.Please display p-values that are less than 0.001 as '< 0.001'.

The author’s answer：

Thank you very much for your suggestion. I have followed your advice and now display the results with p-values less than 0.001 as '< 0.001'.

7. Please ensure that percentages are displayed with one decimal place, and p-values are presented with three decimal places.

The author’s answer：

Thank you for your valuable suggestion. I have now adjusted the presentation to ensure that percentages are displayed with one decimal place, and p-values are presented with three decimal places, as per your recommendation. I appreciate your guidance.

Thank you once again for your valuable review and constructive feedback. It has been immensely helpful for our research. We look forward to receiving further feedback from you.

---

## [Decision Letter · Decision Letter 1]

23 Jan 2024

PONE-D-23-15805R1Delving into Female Breast Cancer: Distinct Disease-Specific Survival outcomes Between Invasive Lobular and Ductal Carcinomas revealed by Propensity Score MatchingPLOS ONE

Dear Dr. Li,

Thank you for submitting your manuscript to PLOS ONE. After careful consideration, we feel that it has merit but does not fully meet PLOS ONE’s publication criteria as it currently stands. Therefore, we invite you to submit a revised version of the manuscript that addresses the points raised during the review process.

We look forward to receiving your revised manuscript.

Kind regards,

Jianhong Zhou

Staff Editor

PLOS ONE

Journal Requirements:

Reviewers' comments:

Reviewer's Responses to Questions

**Comments to the Author**

1. If the authors have adequately addressed your comments raised in a previous round of review and you feel that this manuscript is now acceptable for publication, you may indicate that here to bypass the “Comments to the Author” section, enter your conflict of interest statement in the “Confidential to Editor” section, and submit your "Accept" recommendation.

Reviewer #2: All comments have been addressed

Reviewer #3: All comments have been addressed

2. Is the manuscript technically sound, and do the data support the conclusions?

Reviewer #2: Yes

Reviewer #3: Yes

3. Has the statistical analysis been performed appropriately and rigorously? 

Reviewer #2: Yes

Reviewer #3: Yes

4. Have the authors made all data underlying the findings in their manuscript fully available?

Reviewer #2: Yes

Reviewer #3: Yes

5. Is the manuscript presented in an intelligible fashion and written in standard English?

Reviewer #2: Yes

Reviewer #3: Yes

6. Review Comments to the Author

Reviewer #2: (No Response)

Reviewer #3: The investigation into disease-specific survival (DSS) differences between propensity score-matched invasive ductal carcinoma (IDC) and invasive lobular carcinoma (ILC) patients yielded a significant finding—IDC patients exhibited notably better DSS than their ILC counterparts. Despite the manuscript's well-crafted presentation and thoughtful revisions in response to reviewer feedback, concerns about originality have been raised, given the saturation of similar topics in existing literature. Nevertheless, the study remains valuable and informative for readers.

Here are the requested revisions:

(1) Typically, breast cancer grades range from I to III. Could the authors clarify the classification of Grade IV?

(2) The estrogen receptor (ER) and progesterone receptor (PR) positive rates in this study appear exceptionally high compared to other reports. Please provide a clear definition of ER and PR positivity in the 'Materials and Methods' section.

(3) In Table 1, 'distant stage' and 'AJCC IV' seem to convey the same concept, yet the numerical values differ. Could the authors explain this disparity?

(4) Given the significance of axillary lymph node (LN) metastasis as a prognostic factor in breast cancer, please include information on axillary LN metastasis in both Table 1 and Table 2.

(5) I have indicated a minor revision in the manuscript; please incorporate the suggested changes for improved clarity.

7. PLOS authors have the option to publish the peer review history of their article (what does this mean?). If published, this will include your full peer review and any attached files.

Reviewer #2: No

Reviewer #3: **Yes: **Hyun Jo Youn

---

## [Author Response · Author response to Decision Letter 1]

29 Jan 2024

Dear Editor and Review Experts,

We sincerely appreciate your valuable time and expertise dedicated to reviewing our research. Your review plays a pivotal role in refining and elevating the quality of our study.We have carefully considered your comments and made corresponding adjustments,anticipating that our responses and modifications align with your expectations. Once again, we thank you for your thoughtful review and insightful suggestions. 

Below, we have responded in detail to each review comment:

Reviewer #3

The investigation into disease-specific survival (DSS) differences between propensity score-matched invasive ductal carcinoma (IDC) and invasive lobular carcinoma (ILC) patients yielded a significant finding—IDC patients exhibited notably better DSS than their ILC counterparts. Despite the manuscript's well-crafted presentation and thoughtful revisions in response to reviewer feedback, concerns about originality have been raised, given the saturation of similar topics in existing literature. Nevertheless, the study remains valuable and informative for readers.

Here are the requested revisions:

(1)Typically, breast cancer grades range from I to III. Could the authors clarify the classification of Grade IV?

The author’s answer：We appreciate your thoughtful inquiry regarding the classification of Grade IV in breast cancer grading. As you rightly pointed out, breast cancer grades typically range from I to III, which aligns with the Nottingham or Bloom-Richardson (BR) Score/Grade widely recognized in the field.Additionally, the SEER database documents three distinct systems or formats for describing tumor grading: the 2, 3, or 4 Grade System. The interconversion relationships between these systems can be found on the official SEER database website. In our study, we employ the four-grade system, where Grade IV is defined as "also called undifferentiated or anaplastic.[ Liu D, Wu J, Lin C, Andriani L, Ding S, Shen K, Zhu L. Breast Subtypes and Prognosis of Breast Cancer Patients With Initial Bone Metastasis: A Population-Based Study. Front Oncol. 2020 Dec 2;10:580112. doi: 10.3389/fonc.2020.580112. PMID: 33344236; PMCID: PMC7739957.] we will ensure a comprehensive explanation of Grade IV in our manuscript to enhance clarity.Your keen observation contributes significantly to the refinement of our work, and we are grateful for your valuable input.

(2)The estrogen receptor (ER) and progesterone receptor (PR) positive rates in this study appear exceptionally high compared to other reports. Please provide a clear definition of ER and PR positivity in the 'Materials and Methods' section.

The author’s answer：We sincerely appreciate your meticulous observation and professional review. In response to your query, we conducted a thorough analysis of ER and PR positivity rates based on the original data of 510,385 patients from the SEER Research Plus Data 8 Registries, as illustrated in Figure 1. It's important to note that the dataset contained a significant number of Borderline/Unknown and Recode not available entries.To ensure the integrity of our subsequent analyses, we opted to exclude data with unclear ER and PR statuses, as well as other instances of indeterminate clinical characteristics.

As per the SEER database guidelines, ER and PR positivity is defined as follows: If 1% or more cells stain positive, the test results are considered positive;if less than 1% of cells stain positive, the results are considered negative.We have taken your valuable suggestion into consideration and will incorporate this precise definition into the 'Materials and Methods' section of our manuscript to provide transparency and clarity.We express our gratitude for your professionalism and thoroughness in reviewing our work. 

Figure1: Original distribution of ER and PR status in 510,385 patients with breast cancer.

(3)In Table 1, 'distant stage' and 'AJCC IV' seem to convey the same concept, yet the numerical values differ. Could the authors explain this disparity?

The author’s answer：Thank you very much for your meticulous observation and insightful feedback. Summary staging is the most basic way of categorizing how far a cancer has spread from its point of origin. Summary staging has also been called General Staging, California Staging, and SEER Staging. Summary Staging uses all information available in the medical record, clinical, and pathological. It is frequently used by tumor registries, but not always understood by physicians.According to the SEER Summary Staging Manual 2000-Breast and Female Genital System (https://seer.cancer.gov/tools/ssm/ssm2000/), "Distant" involves distant sites/lymph nodes, further contiguous extension, and metastasis to specific organs or locations. The description of “Distant” is as follows:

Distant site(s)/lymph node(s) involved

Distant lymph node(s):

Cervical, NOS

Contralateral/bilateral axillary

Contralateral/bilateral internal mammary (parasternal)

Supraclavicular (transverse cervical)

Other distant lymph node(s)

Further contiguous extension:

Skin over:

Axilla

Contralateral (opposite) breast

Sternum

Upper abdomen

Metastasis:

Adrenal (suprarenal) gland

Bone other than adjacent rib

Contralateral (opposite) breast - if stated as metastatic

Lung

Ovary

Satellite nodule(s) in skin other than primary breast

Upon a thorough examination of our original data and consideration of the aforementioned description, 'distant stage' appears analogous to 'AJCC IV,' yet there are distinctions. Among the 2457 'distant stage' cases, 2353 align with 'AJCC IV,' while an additional 104 cases belong to the AJCC IIIC classification, specifically involving patients with lymph node N3c staging. We sincerely appreciate your attention to detail and the invaluable insights you have provided,and we hope this clarification addresses the observed disparities.Thank you once again for your valuable contributions to our research.

(4)Given the significance of axillary lymph node (LN) metastasis as a prognostic factor in breast cancer, please include information on axillary LN metastasis in both Table 1 and Table 2.

The author’s answer：Thank you for your valuable advice.I attach great importance to your suggestion regarding the inclusion of axillary LN metastasis. At the data collection stage, we initially considered axillary LN as an independent variable. However, during further analysis, we observed a significant collinear relationship with other factors, which could introduce instability and bias into the results. In order to ensure the accuracy and reliability of the study, we made the decision not to include this variable in the analysis and opted for the comprehensive AJCC stage as a suitable surrogate.

While this particular data did not make it into the final analysis, I am fully cognizant of the pivotal role axillary status plays in breast cancer prognosis.I will continue to follow the research progress in this area and we are collecting local data and taking your suggestion to fully consider axillary status in future studies. Further refine data collection and analysis to validate our results. To ensure the accuracy and reliability of the study.

(5)I have indicated a minor revision in the manuscript; please incorporate the suggested changes for improved clarity.

The author’s answer：Thank you once more for your professional review. I will enhance the manuscript based on your suggestions to improve its overall clarity.

Yours sincerely,

Qingxia Li

Professor, Director

Tel:+86-311-85988779

Email: lqx73@163.com

---

## [Decision Letter · Decision Letter 2]

22 Feb 2024

Delving into Female Breast Cancer: Distinct Disease-Specific Survival outcomes Between Invasive Lobular and Ductal Carcinomas revealed by Propensity Score Matching

PONE-D-23-15805R2

Dear Dr. Li,

We’re pleased to inform you that your manuscript has been judged scientifically suitable for publication and will be formally accepted for publication once it meets all outstanding technical requirements.

Kind regards,

Daniele Ugo Tari, M.D.

Academic Editor

PLOS ONE

Additional Editor Comments (optional):

Dear Authors,

I received the manuscript while the review process was already underway. Despite my repeated attempts to solicit reevaluation from the previous reviewers yielding no response, and in consideration of the time elapsed since submission and your inquiry regarding the delay, along with the completion of two review sessions, I deemed it appropriate to undertake the evaluation of the manuscript myself as a reviewer.

I think that the paper has been improved from the first submission and that all the reviewers' concern has been addressed.

Consequently, I think that it can be accepted in present form.

Sincerely,

Reviewers' comments:

Reviewer's Responses to Questions

**Comments to the Author**

1. If the authors have adequately addressed your comments raised in a previous round of review and you feel that this manuscript is now acceptable for publication, you may indicate that here to bypass the “Comments to the Author” section, enter your conflict of interest statement in the “Confidential to Editor” section, and submit your "Accept" recommendation.

Reviewer #4: All comments have been addressed

2. Is the manuscript technically sound, and do the data support the conclusions?

Reviewer #4: Yes

3. Has the statistical analysis been performed appropriately and rigorously? 

Reviewer #4: Yes

4. Have the authors made all data underlying the findings in their manuscript fully available?

Reviewer #4: Yes

5. Is the manuscript presented in an intelligible fashion and written in standard English?

Reviewer #4: Yes

6. Review Comments to the Author

Reviewer #4: (No Response)

7. PLOS authors have the option to publish the peer review history of their article (what does this mean?). If published, this will include your full peer review and any attached files.

Reviewer #4: No

---

## [Editor Report · Acceptance letter]

20 Mar 2024

PONE-D-23-15805R2 

PLOS ONE

Dear Dr. Li, 

I'm pleased to inform you that your manuscript has been deemed suitable for publication in PLOS ONE. Congratulations! Your manuscript is now being handed over to our production team.

Kind regards, 

on behalf of

Dr. Daniele Ugo Tari 

Academic Editor

PLOS ONE